# Eliciting Compatible Demonstrations for Multi-Human Imitation Learning

**Kanishk Gandhi, Siddharth Karamcheti, Madeline Liao, Dorsa Sadigh**
Department of Computer Science, Stanford University
{kanishk.gandhi, skaramcheti, madelineliao, dorsa}@stanford.edu

**Abstract:** Imitation learning from human-provided demonstrations is a strong approach for learning policies for robot manipulation. While the ideal dataset for imitation learning is homogenous and low-variance – reflecting a single, optimal method for performing a task – natural human behavior has a great deal of *heterogeneity*, with several optimal ways to demonstrate a task. This multimodality is inconsequential to human users, with task variations manifesting as subconscious choices; for example, reaching *down, then across* to grasp an object, versus reaching *across, then down*. Yet, this mismatch presents a problem for interactive imitation learning, where sequences of users improve on a policy by iteratively collecting new, possibly conflicting demonstrations. To combat this problem of demonstrator incompatibility, this work designs an approach for 1) *measuring the compatibility* of a new demonstration given a base policy, and 2) *actively eliciting more compatible demonstrations* from new users. Across two simulation tasks requiring long-horizon, dexterous manipulation and a real-world "food plating" task with a Franka Emika Panda arm, we show that we can both identify incompatible demonstrations via post-hoc filtering, and apply our compatibility measure to actively elicit compatible demonstrations from new users, leading to improved task success rates across simulated and real environments.[1]

**Keywords:** Interactive Imitation Learning, Active Demonstration Elicitation, Human-Robot Interaction

## 1 Introduction

Interactive imitation learning [1, 2, 3] from a pool of human demonstrators is a scalable approach for learning multi-task policies for robotic manipulation [4, 5, 6]. Yet, such approaches have a critical problem, especially in the low-to-moderate data regime: data from multiple human demonstrators often have *conflicting* modes, where two users provide opposing behaviors for a single task – behaviors that manifest as subconscious, random choices. For example, consider the nut-on-peg task in Fig. 1: one user (in orange) approaches the nut by moving *across the table, then down*, while the other user (blue) reaches *down, then across*.

Training on aggregated batches of data in series – starting with a base policy, adding small amounts of data from new users, and retraining the policy after each batch – is common in interactive imitation formulations [2, 3]; unfortunately, when we add a small number of conflicting demonstrations during the interaction phase, the retrained policy attempts to cover both the base demonstrations *and* the new set. This leads to incongruent overfitting, where a policy – even one equipped to learn multimodal behaviors [7, 8, 9] – tries to fit the base set for most of a trajectory, but overfits to the new set for a small subset of the state space, often with catastrophic failure modes.

To mitigate this problem, this work tackles two questions: 1) *how can we measure the compatibility between a new demonstrator and an existing policy*, and 2) *how can we use this measure to actively elicit better demonstrations from a new user?*

While our approach for measuring and eliciting compatible demonstrations *during online collection* is novel, prior work has studied the impact of suboptimal demonstrations on learning. Most relevant, Mandlekar et al. [10] introduce RoboMimic, a suite of simulated manipulation tasks that consist

---

[1]Additional videos & results: https://sites.google.com/view/eliciting-demos-corl22/home

of demonstrations collected from 6 humans of varying qualities (Worse, Okay, and Better). This work and recent followups [11] show how policies degrade in the presence of heterogenous data with multimodal behaviors, finding that training imitation learning policies on data from a single demonstrator can often exceed the performance of training on equivalent amounts of data from *multiple* demonstrators. A related body of work on offline policy learning propose approaches for overcoming suboptimality by using extra annotations, usually in the form of rewards or returns for each demonstration. These methods seek to reweight [12, 13, 14], or directly filter out demonstrations [15, 16, 17]. Yet, these are all post-hoc approaches that operate *after* demonstrations have been collected. None of these prior approaches target the root cause: informing users of the compatibility of their demonstrations, *as* they collect data in the first place.

In contrast, our proposed approach targets this root cause by first *learning* a fine-grained measure of demonstrator compatibility subject to an initial base policy, and then *operationalizing* this measure to inform users *while* they collect demonstrations. Our compatibility measure is derived by sorting individual state-action pairs in the data based on their likelihood and novelty under the base policy. By plotting each state-action along a 2D "map" with these two properties as the axes, we are able to learn visual boundaries denoting "aligned and compatible" demonstrations (those with high likelihood and low novelty) to those that are grossly incompatible (low likelihood and low novelty) – an approach inspired by prior work in dataset interpretability [18, 19]. From these maps, we identify a scoring function and set of thresholds to define our demonstrator compatibility measure, which we then cheaply and efficiently deploy during active data collection. Crucially, with this fine-grained measure, users with "incompatible" demonstrations are given rich feedback as to *where* in their demonstration the incompatibility arose coupled with visualizations of compatible demonstrations from the base policy's training set, guiding their behavior moving forward.

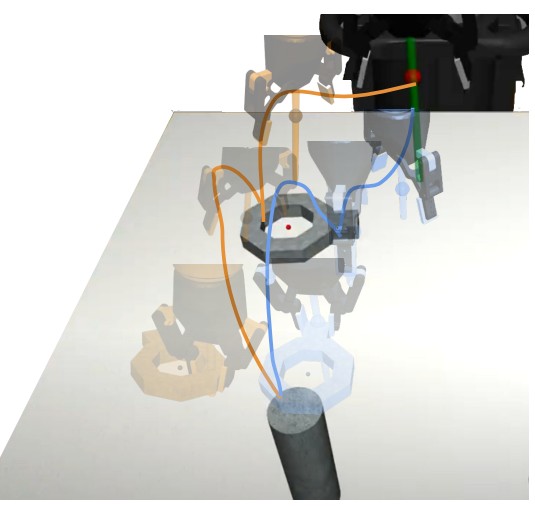

Figure 1: Visualization of two users (orange and blue) inserting a nut onto a round peg. Both demonstrators approach the round nut, pick it up and insert it into the peg in very different ways, introducing *heterogenous* data that is incompatible with the base policy, hurting performance when retrained.

Our results across two simulated tasks from prior work [10, 20] and a real-world "food-plating" task with a Franka Emika Panda arm show that our compatibility measure enables the elicitation of more compatible demonstrations – demonstrations that, when added to the base dataset, lead to a boost of up to 25% in success rate relative to a 20% decrease in performance when collecting data naively, without feedback.

## 2 Related Work

**Identifying & Learning from Diverse Demonstrations.** Our definition of *diversity* subsumes both *heterogeneity* – where demonstrators capture multiple distinct, but equally optimal behaviors – and *suboptimality* – where users provide demonstrations that are poor relative to some objective reward function. On examining the effect of learning from these diverse demonstrations, Mandlekar et al. [10] found that imitation learning models trained on demonstrations from a single "proficient" human achieved higher success rates than models trained on significantly larger dataset sourced from a mixture of human demonstrators with varying optimality. Gopalan et al. [21] further show that without prompting users with videos or specific subtasks, only a *third* of demonstrators provide compatible demonstrations. These works motivate our approach by showing that in naive demonstration collection, heterogeneity and suboptimality are prevalent; instead, we need a direct way to intervene during collection, actively inform users, and guide them to provide compatible demonstrations.

Other work studies *learning* from diverse demonstrations. Several approaches [7, 8, 9, 22, 23] model the multimodality of diverse demonstrations directly, usually by explicitly handling each mode of the data. Unfortunately, these approaches fail in the sequential, interactive imitation learning setting that we study, where we gradually introduce new modes and retrain after each collection. A separate line of work tackles learning with suboptimal data through the use of offline reinforcement learning [12, 13, 14], filtered imitation [11, 15, 16, 17], or inverse reinforcement learning [24, 25, 26]. These methods use extra reward annotations for each demonstration, and generally try to reweight or selectively remove suboptimal data. Unlike these approaches, this work tackles the root cause of the problem – how to collect compatible demonstrations from humans in the first place, removing the need to filter out 10s – 100s of painstakingly collected demonstrations.

**Interactive Imitation Learning.** Most of the work in interactive imitation learning attempts to address the problem of *covariate shift* endemic to policies trained via behavioral cloning. Ross et al. [1] introduce DAgger, a method for iteratively collecting demonstrations by relabeling the states visited by a policy interactively, teaching the policy to recover. Later work builds on DAgger, primarily focusing on reducing the number of expert labels requested during training. These variants identify different measures such as safety and uncertainty to use to query the expert limiting the amount of supervision a user needs to provide [2, 20, 27, 28, 29, 30]. The focus of these works is on learning from a *single* user rather than from *multiple* users demonstrating heterogenous behaviors.

Our proposed approach presents a novel interactive imitation learning method that learns to actively elicit demonstrations from a pool of *multiple users during the course of data collection*, with the express goal of guiding users towards compatible, optimal demonstrations.

## 3  Problem Setting

We consider sequential decision making tasks, modeled as a Markov Decision Process (MDP) with the following components: the state space $\mathcal{S}$, action space $\mathcal{A}$, transition function $\mathcal{T}$, reward function $R$, and discount factor $\gamma$. In this work, we assume sparse rewards only provided on task completion. A state $s = (o_{\text{grounded}}, o_{\text{proprio}}) \in \mathcal{S}$ is comprised of a grounded observation (either coordinates/poses of objects in the environment, or an RGB visual observation) and the robot's proprioceptive state. An action $a \in \mathcal{A}$ is a continuous vector of robot joint actions. We assume access to a small dataset of trajectories $\mathcal{D}_{\text{base}} = \{\tau_1, \tau_2 \ldots \tau_N\}$ where each trajectory $\tau_i$ is a sequence of states and actions, $\tau_i = \{(s_1, a_1) \ldots (s_T, a_T)\}$. We train an initial policy $\pi_{\text{base}}$ on this dataset via behavioral cloning.

Our approach has two components: 1) developing a measure of the compatibility of a new demonstration with an existing base policy, and 2) building a method for actively eliciting compatible demonstrations from users. For the first component, we learn a measure $\mathcal{M}(\mathcal{D}_{\text{base}}, (s_{\text{new}}, a_{\text{new}})) \in [0, 1]$ that defines a compatibility at the granuality of a state-action pair. For the second component, we use our fine-grained measure $\mathcal{M}$ to provide rich feedback to users about their demonstrations, in addition to deciding whether to accept/reject a new demonstration $\tau_{\text{new}}$. The set of new demonstrations the user collects comprises $\mathcal{D}_{\text{new}}$ – after each collection step, we train a new policy $\pi_{\text{new}}$ on $\mathcal{D}_{\text{base}} \cup \mathcal{D}_{\text{new}}$.

Under our definition of compatibility (and the measure $\mathcal{M}$ we derive), we hope to guide users to provide new datasets $\mathcal{D}_{\text{new}}$ such that $\pi_{\text{new}}$ will have as high of an expected return as possible, and minimally, a higher expected return than $\pi_{\text{base}}$. In other words, our definition of compatibility asks that new data should only help, not hurt performance relative to the initial set.

## 4  Learning to Measure Compatibility in Multi-Human Demonstrations

We first derive a general compatibility measure $\mathcal{M}$ given a base set of demonstrations, then evaluate our measure through a series of case studies grounded in real, user-provided demonstration data.

### 4.1  Estimating Compatibility and Identifying Good Demonstrations

An idealized compatibility measure $\mathcal{M}$ has one role – estimating the performance of a policy $\pi_{\text{base}}$ that is retrained on the union of a known base dataset $\mathcal{D}_{\text{base}}$ and a new dataset $\mathcal{D}_{\text{new}}$. Crucially, $\mathcal{M}$ needs to operate at a granular level (ideally at the level of individual state-action pairs), without incurring the cost of retraining and evaluating $\pi_{\text{base}}$ on the new dataset. Phrased this way, there is a clear connection to pool-based active learning [31], informing a choice of a plausible metrics that could help predict downstream success. While many metrics could work, we choose two easy-to-compute metrics that lend themselves well to interpretability: the *likelihood* of actions $a_{\text{new}}$ in $\mathcal{D}_{\text{new}}$ under $\pi_{\text{base}}$, measured

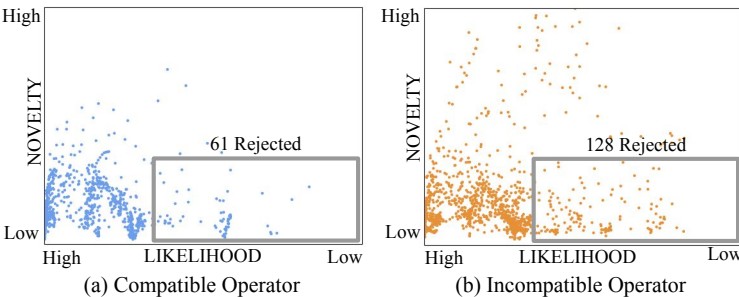

(a) Compatible Operator        (b) Incompatible Operator

Figure 2: Plots, or "maps" of the likelihood and novelty values for demonstrations recorded by the blue *compatible* operator (a), and the orange *incompatible* operator with respect to the base policy $\pi_{\text{base}}$. The gray border indicates a filter with our chosen thresholds $(\lambda, \eta)$. The orange operator performs far more *low-likelihood* actions in familiar states, earning a lower compatibility score.

by the negative mean squared error between actions predicted by $\pi_{\text{base}}(s_{\text{new}})$ and $a_{\text{new}}$, as well as the *novelty* of a given state, measured by the standard deviation in the predicted actions under the base policy. To turn these two metrics into a single measure, we borrow from interpretability and active learning literature [18, 19]; we plot, or "map" on a 2D plane the likelihood and novelty values for each trajectory in our dataset, with the goal of identifying two thresholds $(\lambda, \eta)$ on likelihood and novelty – thresholds that allow us to identify a given trajectory based on its compatibility.

The final piece is setting the thresholds for novelty $(\eta)$ and likelihood $(\lambda)$. While this can be done in many ways – often just by looking at the "mapped" dataset on the 2D plane [18] – in this work, we assume the ability to obtain or collect a handful of *a priori incompatible* demonstrations that we can use to build contrast sets to regress a threshold. We define compatibility $\mathcal{M}(\mathcal{D}_{\text{base}}, (s_{\text{new}}, a_{\text{new}})) \in [0, 1]$ as parameterized by thresholds $(\lambda, \eta)$:

$$\mathcal{M} = \begin{cases} 1 - \min\left(\frac{(\pi_{\text{base}}(s_{\text{new}}) - a_{\text{new}})^2}{\lambda}, 1\right) & \text{if novelty}(s_{\text{new}}) < \eta \\ 1 & \text{otherwise.} \end{cases}$$

A state-action pair is compatible $(\mathcal{M}(\mathcal{D}_{\text{base}}, (s_{\text{new}}, a_{\text{new}})) = 1)$ if the likelihood of $a_{\text{new}}$ under the base policy is high, *or* the novelty of $s_{\text{new}}$ is $> \eta$; incompatibility is defined smoothly on the inverse interval with a state-action pair being completely incompatible if $(\mathcal{M}(\mathcal{D}_{\text{base}}, (s_{\text{new}}, a_{\text{new}})) = 0)$ when $(\pi_{\text{base}}(s_{\text{new}}) - a_{\text{new}})^2 \geq \lambda$. Intuitively, this measure mirrors what one expects when collecting new data – in states where the policy is confident about what actions to take, reject other actions, and in new states where the policy is uncertain, more, diverse data is good. The following subsection validates this measure against demonstration data collected by real users on a series of tasks, showing how we can use $\mathcal{M}$ as a filter to boost policy performance in the presence of a large dataset of heterogenous demonstrations.

## 4.2   Case Studies and Experiments

To validate our measure $\mathcal{M}$, we consider three tasks in simulation, each with a set of demonstrations collected by multiple operators. For each task, we choose one operator to initialize the base set of demos $\mathcal{D}_{\text{base}}$ and use trajectories from another operator to form $\mathcal{D}_{\text{new}}$. For a more detailed summary of the tasks datasets, and policy training procedure, please consult the Supplementary Material.

**Square Nut** [10]: The goal of the task is to place a square nut onto a square peg. We use the environment and demonstrations collected by Mandlekar et al. [10], consisting of 200 demonstrations from a proficient operator, and 50 demonstrations each from 6 operators of varying qualities. We initialize $\mathcal{D}_{\text{base}}$ with 50 demonstrations from the proficient operator, and use demonstrations from each of the 4 more efficient operators as the different $\mathcal{D}_{\text{new}}$.

**Round Nut** [10, 20]: Similar to the prior task, the goal is to place a *round* nut onto a *round* peg (as in Fig. 1). For this task, we collected 30 demonstrations from a proficient operator and 30 demonstrations each from 3 other operators – where the data provided for one operator is taken

| Operator | Square Nut | | Round Nut | | Hammer Placement | |
|---|---|---|---|---|---|---|
| | Naive | $\mathcal{M}$-Filtered | Naive | $\mathcal{M}$-Filtered | Naive | $\mathcal{M}$-Filtered |
| Base Operator | 38.7 (2.1) | - | 13.3 (2.3) | - | 24.7 (6.1) | - |
| Operator 1 | 54.3 (1.5) | 61.0 (4.4) | 26.7 (11.7) | 32.0 (12.2) | 38.0 (2.0) | 39.7 (4.6) |
| Operator 2 | 40.3 (5.1) | 42.0 (2.0) | 22.0 (7.2) | 26.7 (5.0) | 33.3 (3.1) | 32.7 (6.4) |
| Operator 3 | 37.3 (2.1) | 42.7 (0.6) | 17.3 (4.6) | 18.0 (13.9) | 8.0 (0.0) | 12.0 (0.0) |
| Operator 4 | 27.3 (3.5) | 37.3 (2.1) | 7.3 (4.6) | 13.3 (1.2) | 4.0 (0.0) | 4.0 (0.0) |

Table 1: Success rates (mean/std across 3 training runs) for policies trained on $\mathcal{D}_{\text{new}}$ from different operators. $\mathcal{M}$-Filtered denotes filtering by compatibility scores.

directly from Hoque et al. [20]. We initialize $\mathcal{D}_{\text{base}}$ using the proficient operator's demonstrations, and we randomly sample 10 demonstrations from each operator to get different $\mathcal{D}_{\text{new}}$.

**Hammer Placement** [32]: Here the goal is to open a drawer and place a hammer inside (see Fig. 3). We collected 25 demonstrations from a proficient operator to initialize the $\mathcal{D}_{\text{base}}$ and 5 demonstrations from each of 4 other operators to get different $\mathcal{D}_{\text{new}}$.

**Policy Architecture, Training, and Evaluation.** We train an ensemble of 5 MLP policies using coordinate-based state observations and the robot's proprioceptive state to predict 7-DoF joint actions. Models are trained for 1000 epochs, and evaluated every 200 epochs; we select our policy by taking the checkpoint with the highest success rate averaged over 50 rollouts.

**Case Study: Naïve Training.** To establish baseline performance, we compare the performance of policies trained on different $\mathcal{D}_{\text{new}}$ given by different operators to identify meaningful markers. The results in Table 1 show a large difference in the success rate of $\pi_{\text{new}}$ depending on which operator provided $\mathcal{D}_{\text{new}}$. For example, on Square Nut, sampling $\mathcal{D}_{\text{new}}$ from Operator 1 (the same operator who provided $\mathcal{D}_{\text{base}}$) leads to an *improvement of 15.6%* over the base policy, while sampling from Operator 4 leads to a *reduction of 11.4%*. Similarly, Operator 1 (again, the operator who provided $\mathcal{D}_{\text{base}}$) in Round Nut (the blue line, Fig. 1) almost *doubles* the performance of the base policy while Operator 4 (orange line in Fig. 1) *halves* it. In general, training on data with incompatible demonstrations can degrade policy performance by seemingly arbitrary amounts.

**Experiment: Compatibility-Driven Filtering.** Finally, we evaluate the ability of our measure $\mathcal{M}$ to identify compatible demonstrations from fixed pools of demonstration data. We first note the result of "mapping" the various demonstrators to provide some additional visual intuition, grounded in the case study results; Fig. 2 shows the 2D "maps" for Operator 1 (orange) and Operator 4 (blue). These plots (see the Supplemental Material for plots of all operators), present a key insight: *incompatible operators provide more unlikely actions for familiar, or less-novel, states*. Closing the loop and using our measure $\mathcal{M}$ to filter out incompatible demonstrations, we see strong results: Table 1 shows that removing incompatible samples leads to performance *that is at least as good, if not much better than the base policy*. For instance, we see an improvement of 6.7% for Operator 1 (the most compatible), on the Square Nut task and of 10% for Operator 4 (the least compatible).

## 5 Actively Eliciting Compatible Demonstrations

Our pipeline starts traditionally: operators are provided a task specification (e.g., "place the square nut on the square peg") and are given three episodes of practice with the robot. We then enter our active elicitation process (see Fig. 3 for a full overview of our interface). First, is *prompting*; each operator is shown 5 rollouts from $\mathcal{D}_{\text{base}}$ to understand the style of the base operator. Second is *demonstration*, where the operator is tasked with providing a demonstration similar to that which they were just shown. Critically, as they record their demonstration, we use our measure $\mathcal{M}$ to provide online feedback – a green indicator for compatible actions, and a red indicator for incompatible actions.

After demonstrating, begins the *feedback* phase; if the number of incompatible state-action pairs – $(s_{\text{new}}, a_{\text{new}})$ pairs with $\mathcal{M}(\mathcal{D}_{\text{base}}, (s_{\text{new}}, a_{\text{new}})) = 0$ – exceeds 5% of the accepted demonstration length, then the demonstration is rejected. Rather than a binary decision, we take the opportunity to provide rich feedback: using a simple sliding window, we identify 3 candidates with the *highest average incompatibility* from the rejected demonstration. We show this to the user, along with a demonstration from $\mathcal{D}_{\text{base}}$ retrieved based on state-similarity to the 3 candidates. With this contrastive

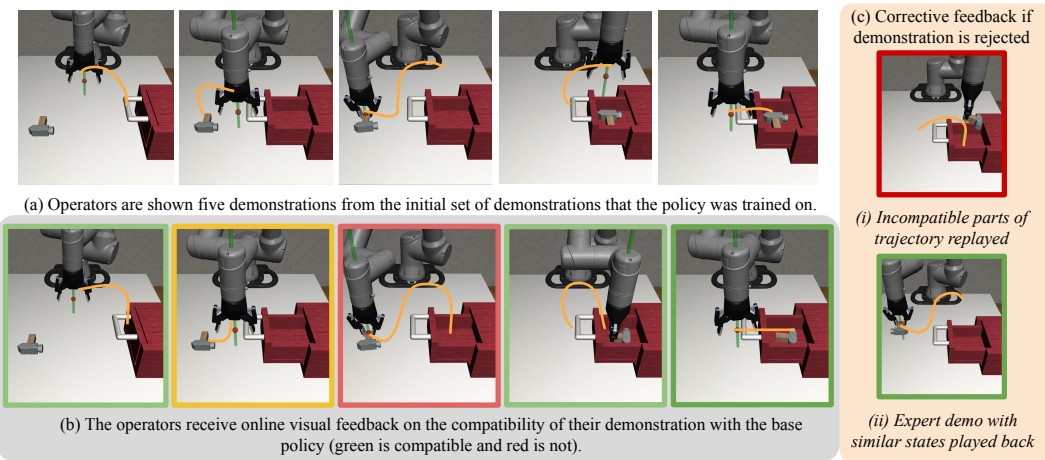

(a) Operators are shown five demonstrations from the initial set of demonstrations that the policy was trained on.

(b) The operators receive online visual feedback on the compatibility of their demonstration with the base policy (green is compatible and red is not).

(c) Corrective feedback if demonstration is rejected

*(i) Incompatible parts of trajectory replayed*

*(ii) Expert demo with similar states played back*

Figure 3: The three phases of our active elicitation interface spanning the initial *prompting* phase (a), the subsequent *demonstration* phase with live feedback (b), and finally, the *feedback* phase (c).

feedback, we find that users are able to more effectively pinpoint *where* in the demonstration they went wrong, and learn how to provide more compatible demonstrations moving forward. We continue this process until a desired number of demonstrations have been recorded. Finally, $\pi_{\text{base}}$ is updated to $\pi_{\text{new}}$ every time a batch of demonstrations is collected from a demonstrator.

## 5.1 User Studies – Simulation

We build a browser interface for users to provide teleoperated demonstrations for two tasks in simulation: `Round Nut` and `Hammer Placement`. We conduct user studies in two conditions, *naive* and *informed*. Under the naive condition, users get to practice the controls during 3 practice episodes, before collection demonstrations *without any additional prompting*. Under the informed condition, users are provided the active elicitation interface above.

**Round Nut.** $\mathcal{D}_{\text{base}}$ consists of 30 trajectories collected by a proficient operator. The users are randomly assigned to one of the two conditions, and are asked to collect 10 successful demonstrations $\mathcal{D}_{\text{new}}$. We conduct user studies for this task with $n = 16$ participants.

**Hammer Placement.** $\mathcal{D}_{\text{base}}$ consists of 25 trajectories collected by a proficient operator. For this task, we perform a longitudinal study with $n = 4$ participants, where users are first asked to collect 5 demonstrations in the naive condition and then collect 5 demonstrations in the informed condition. This allows us to measure the effect of the interface in eliciting demonstrations within subjects.

**Results**. For both tasks, we find that policies trained on demonstrations collected via active elicitation consistently outperform those from the naive condition. Our studies indicate that naive participants find it difficult to provide compatible demonstrations, worsening the base policy on average. In contrast, informed demonstrators improve the base policy, beating the naive condition by 6.1% on `Round Nut` and 11% on `Hammer Placement`. From our longitudinal studies on the latter, we find that informed demonstrations beat naive demonstrations from the same user for 3/4 operators, clearly indicating that active elicitation is a strong approach for interactive imitation. We also find that the length of trajectories recorded by informed demonstrators is closer to those of the base demonstrator (see Supplementary). To test if the high frequency feedback from our interface discouraged users from providing corrections, we also perform experiments with collecting interactive, human-gated (Hg-DAgger [2]) demonstrations. Informed data collection outperforms naive data collection for 3 out of 4 users even in this setting and informed users intervene at about the same rate as informed ones (see supplementary for details).

## 5.2 User Studies – Real Robot

**Food Plating.** As depicted in Fig. 4, the task requires the Franka Emika Panda arm to lift the pink pan from a "stove" and guide it over a plate, where it is responsible for transferring a fried egg

| Task | Base | Naïve | Naïve + Filtered | Informed |
|---|---|---|---|---|
| **Round Nut** | 13.3 (2.3) | 9.6 (4.6) | 9.7 (4.2) | 15.7 (6.0) |
| **Hammer Placement** | 24.7 (6.1) | 20.8 (15.7) | 22.0 (15.5) | 31.8 (16.3) |
| **[Real] Food Plating** | 60.0 | 30.0 (17.3) | - | 85.0 (9.6) |

Table 2: Success rates (mean/std across users) for the user studies evaluating both naive and informed demonstration collection against base users.

from the pan to the plate without dropping it on the table. The location of the plate varies across episodes, and the robot has to learn to move and tilt the pan in a way that the egg gently slides onto the plate. $\mathcal{D}_{\text{base}}$ consists of 15 demonstrations collected by a single, proficient operator that maintains a consistent strategy of lifting the pan and tilting it forward ("vertically plating") to transfer the egg.

**Policy.** We train an ensemble of 5 policies that predict 7-DoF joint actions from visual observations (RGB images) and proprioceptive states. We use a ResNet-34 backbone pretrained on ImageNet [33] to encode the visual observations, followed by a single MLP to predict actions from the concatenated ResNet embedding and proprioceptive state (see the Supplementary Material for additional details). We select our final model based on the best validation loss (MSE) after 20 epochs of training.

**Experimental Setup.** We build an analog of our browser interface for active elicitation in a console UI. Unlike simulation, users provide demonstrations *kinesthetically* by physically moving the robot arm. Before collecting demonstrations, users are time to explore the controls of the robot and practice moving it. We conduct a within-subjects study with $n = 4$ users who first perform the naive collection of 5 demonstrations, then collect 5 demonstrations using the active elicitation interface.

**Results.** Training on demonstrations collected with active elicitation outperform the base policy by 25% and the naive condition by 55%. Additionally, similar to simulation, we see that naive demonstration collection often hurts policy performance.

## 6   Discussion

Driving this work has been a simple story; in §4 we performed a thorough analysis and set of experiments on *pre-existing datasets* to show that heterogeneity is a very real problem in multi-human imitation learning datasets, one that can harm policy performance in arbitrary ways. Through this section, building up to §4.1 we were able to profile these existing datasets, and regress a straightforward compatibility measure $\mathcal{M}$ that hinged on two attainable features of the base policy – the likelihood and novelty of a new state-action pair under the existing model parameters. Given these case studies and our proposed measure, we turned to active elicitation in §5, building an interface to guide users towards producing more compatible demonstrations, as depicted in Fig. 3.

On our simulated tasks (Table 2), we showed that an uninformed collection procedure – no prompting, no fine-grained feedback – leads to unequivocally *worse* performance than the base policy downstream, due to the introduced heterogeneous data. However, with our active elicitation and prompting behavior, we see the *opposite* – more data leads to significant improvements over the base policy, by a broad margin. This story is only further confirmed with our experiments on a real robot, as we had users provide demonstrations kinesthetically, showing more extreme results. On the seemingly simple "food plating" task, uninformed demonstration collection led to a broad spectrum of behaviors (e.g., the sideways plating depicted in Fig. 4 vs. the idealized vertical plating of the base policy) driving the success rate down almost 30% relative to the base policy. In contrast, the same users – with just 1-2 episodes of feedback – were able to learn to provide compatible demonstrations, with their "informed" demonstrations leading to an absolute policy success rate of 85%. Not only is the active elicitation procedure highly effective, but it is fundamentally cheap, generalizable across various control schemes (e.g., the discrete keyboard control of simulation vs. the kinesthetic teaching in the real world), and is broadly applicable across various types of manipulation tasks.

**Limitations.** We make several assumptions in this work that may impede future applications. Our biggest assumption is that the preferences reflected in the initial dataset are the "ground-truth" to aspire to. In settings where users *have specific, conflicting preferences about how to accomplish a task*, our approach falls short. Consider the age-old question of when to pour milk into a cereal bowl – namely, before or after the cereal has made it to the bowl. Under our approach, given a base dataset

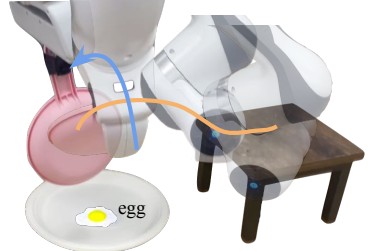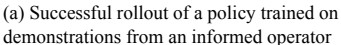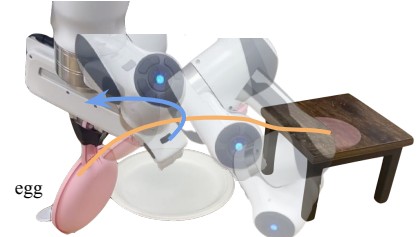

(a) Successful rollout of a policy trained on demonstrations from an informed operator

(b) Failed rollout of a policy trained on demonstrations from a naive operator

Figure 4: Evaluation trajectories for the Informed (a) & Naive (b) conditions. In (b), the user provides conflicting demos (a "sideways tilt," moving laterally and rotating sideways) compared to the initial demos ("vertical plating"). The resulting policy moves sideways and forward before failing.

of cereal being poured into the bowl first, every subsequent user trying to teach our system – despite their preferences – will be guided towards a "cereal-first" approach. This paper focuses on tasks that do not have this quality – tasks where the order in which one picks up a nut, or opens a cupboard is inconsequential – but we hope that future work expands on our initial formulation, and allows for cases where users with strong preferences can override the default demonstration set. In cases where users do have strong preferences, we are instead able to allow users to make an informed choice about whether they want to spend the time collection more demonstrations to reflect their preferences. Compared to the status quo where users just don't know if their demonstrations are incompatible in the first place, our approach provides them with more feedback to make an informed choice when it comes to demonstration collection.

The second limitation stems from our assumption of access to the initial dataset of demonstrations for training the base policy. Access to this data is necessary for policy retraining, as well as for developing our compatibility measure $\mathcal{M}$. One could imagine scenarios where (due to privacy or release feasibility) this initial dataset cannot be made available. While we could obtain reasonable proxies to use in our method (e.g., by rolling out several times in an environment to collect a new dataset), performance of the learned policy would likely suffer. Luckily, methods for continual learning and model interpretability – reaching within the black-box – are growing, presenting a starting point for how to proceed in cases where initial datasets are not available.

**Looking Forward.** A pivotal idea of this work is breaking the traditional influence pattern in human-robot interaction, where a human influences a robot (by providing demonstrations, feedback, preferences). There is power in going the *other* way – allowing the robot to guide and prompt humans as to the type of data needed to improve. Future work will build upon our Hg-DAgger results, refining our interface for active elicitation for more responsive interactive imitation algorithms. Other work might consider language, providing users with higher-level feedback ("no, reach *across* first!") for more effective guidance.

## 7   Conclusion

We introduce a two-phase approach for multi-human interactive imitation learning that 1) derives a fine-grained measure of demonstrator compatibility with an existing policy, and 2) develops an interface for actively eliciting compatible demonstrations from new users. Through our user studies on two diverse simulation environments, and a real-world Franka Emika Panda arm, we demonstrate the importance of demonstrator compatibility. We show the inefficiency of naïve, unprompted demonstration collection and more importantly, we show that spending time to prompt users with fine-grained feedback about the demonstrations they provide is extremely useful, boosting the performance of the task policy by up to 25% absolute success rate from just 5 demonstrations.

**Acknowledgments**

This project was sponsored by NSF Awards 19417222, 2132847, and 2218760, the Office of Naval Research (ONR), and the Air Force Office of Scientific Research (AFOSR). Toyota Research Institute ("TRI") additionally provided funded funds to support this work. Siddharth Karamcheti is further grateful to be supported by the Open Philanthropy Project AI Fellowship.

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
