# OpenReview forum: "Eliciting Compatible Demonstrations for Multi-Human Imitation Learning"
_robot-learning.org/CoRL/2022/Conference — CoRL 2022 Poster_

### Official Review · Reviewer_tzdZ · 2022-07-14

**Originality:** Good
**Technical Quality:** Good
**Clarity Of Presentation:** Excellent
**Impact:** 3

**Recommendation:**

Weak Accept: I recommend accepting the paper, but will not argue for my recommendation if the majority of other reviewers have a different opinion.

**Summary:**

This paper presents an approach for measuring and actively soliciting "compatible demonstrations" in interactive imitation learning (IL) from multiple humans, where compatibility is defined relative to the behavior of the initial demonstrator. As opposed to prior work that handles the issue of suboptimal and multimodal humans in IL in a post-hoc manner (after the dataset has been collected), this work proposes to influence the data collection process itself toward the collection of more compatible demonstrations. They define a novel compatibility metric over state-action pairs as a synthesis of (1) action likelihood and (2) state novelty relative to the initial policy, and they apply this metric to both filtering data and actively soliciting compatible data. In user studies in simulation tasks and a real food-plating task, they show that the active method achieves higher success rates over a naive approach in all tasks.

**Issues:**

1. Add more technical details to the text or appendix, especially on the computation of the likelihood metric.
2. Add more justification for the chosen compatibility metric, particularly how to address the high novelty yet incompatible case above.
3. Add a “Prompt-Only” baseline or similar baseline for the experiments with active solicitation.
4. Add more quantitative results from the real robot experiment data.

**Quality Of The Limitations Section:**

Limitations are addressed clearly

**Reviewer Expertise:**

4: The reviewer is confident but not absolutely certain that the evaluation is correct

**Robotics Focus:**

Sufficient demonstration on hardware

**Strengths And Weaknesses:**

This paper tackles an important problem in imitation learning, i.e., that of multimodal and suboptimal human demonstrations, especially in the multiple human setting. The high-level approach is compelling and novel: instead of the conventional approach of processing human data after it has been collected, the robot can actively influence the human to provide the data in a compatible manner. This is certainly a promising and intuitive strategy for mitigating the compatible demonstration problem. The paper is very well-written, clear, and easy to follow, and the figures are good for providing intuition as well. The experiments are convincing and include both real user studies and real robot experiments.

My main reservations are with the technical aspects of the approach. Firstly, although the overall framework of interactively querying for compatible demonstrations is novel, the actual mechanism for doing so is not particularly novel. Specifically, they propose using the likelihood of actions and the novelty of states under the base policy, neither of which is a novel metric. Secondly, the authors do not provide sufficient technical detail. I was not able to find the details of the likelihood and novelty computations in the main text or supplemental material: I’m assuming novelty is computed as the policy output variance across the ensemble of neural networks, but what does “likelihood” mean? Why does it only take the action as input, when likelihood seems to be a state-dependent measure (a particular action can be more likely at a given state than another one)? Furthermore, how exactly are the thresholds $\lambda$ and $\eta$ calculated, and what are the exact values they take in the experiments? Clarifications here would be appreciated. Thirdly, I have some reservations regarding the overall soundness of the approach. (1) In interactive IL, the policy is iteratively improved by new intervention data. Measuring compatibility against the base policy $\pi_{base}$ seems to prevent future policies from significantly outperforming the base policy, which runs counter to the objective of interactive IL. Is the $\pi_{base}$ here iteratively updated to $\pi_{new}$ when computing compatibility in later rounds of interactive learning? (2) Currently, high novelty demonstrations are deemed compatible, but this seems flawed - if multiple operators are providing demonstrations in novel regions of the state space, aren’t they free to provide demonstrations that are incompatible with each other?

Some more notes:
- Line 95: “these approaches fail in the sequential, interactive imitation learning setting that we study.” Is there justification or a source for this claim? I don’t readily see why an approach like Implicit BC wouldn’t work on the aggregated dataset $D_{new}$.
- Line 212: an incompatible state-action pair is said to have $\mathcal{M} = 0$. Isn’t $\mathcal{M}$ a continuous measure that’s unlikely to be exactly 0?
- Line 216: The feedback phase uses state-similarity to retrieve relevant snippets of demonstrations. How is this done in image space? L2 pixel distance?
- The active informed algorithm makes several modifications compared to the naive baseline, but is only compared in a composite manner. It would be good to include some ablations or additional baselines for more convincing experimental results. For instance, experiments with only the prompting phase (showing operators the base demonstrations) to see if the demonstration and feedback phases are helpful.
- The real results section is quite short with only two quantitative numbers reported. It would be good to see more detailed metrics here (e.g., standard deviation or other quantitative results). Also, justification or intuition on why the performance gap is so much larger in the real experiment compared to the simulation experiments.

**Post Revisions:** The reviewers have adequately addressed most of my concerns and I have raised my assessment from Weak Reject to Weak Accept. However, some remaining concerns are that the method lacks algorithmic novelty and the real experimental results are a bit lacking.

**Summary Of Recommendation:**

Originally was a Weak Reject but after revisions I modify my recommendation to a Weak Accept.

---

> ### Author Response · Authors · 2022-08-21
> **Response to Reviewer tzdZ (1/2)**
>
>
> We’re grateful to Reviewer `tzdz` for their detailed read of our paper; we address their concerns around ablations for active elicitation, details of the compatibility metric, the updation of the base policy, and other details of our elicitation procedure.
>
> > The active informed algorithm makes several modifications compared to the naive baseline, but is only compared in a composite manner. It would be good to include some ablations or additional baselines for more convincing experimental results.
>
> This is a very fair criticism! The Metareviewer adds to this pointing out that it isn’t clear why “measure demonstration adequacy isn't sufficient by itself” for collecting new demonstrations from users – why do we need this active elicitation interface?
>
> Adding real-time feedback in addition to prompting make it easier for users to understand which parts of their trajectory were the most compatible. The interface with feedback based on colors is intuitive and comes at almost no additional overhead to judging the compatibility of a trajectory. To test if judging the compatibility of a demonstration is adequate we have **added more experiments to explicitly test this!** We add a naive demonstrator + compatibility filtering condition to our active elicitation section. For this condition, we filter out incompatible demos based on the compatibility score computed using values of $\lambda$ and $\eta$ defined in Section 4. We show the following results:
>
> We find that filtering via our compatibility measure (as suggested by `tzdZ` and the MetaReviewer) shows minor improvements over “naive” but performs significantly worse than “informed” – the active elicitation really does help lower the cost of collecting significantly higher quality demonstrations.
>
> > The real results section is quite short with only two quantitative numbers reported. It would be good to see more detailed metrics
>
> We add additional experimental details and evaluation metrics (like trajectory lengths and operator-wise success rates) to our experimental results that hopefully paint a clearer picture.
>
> > Is Πbase here iteratively updated to πnew when computing compatibility in later rounds of interactive learning?
>
> It indeed is! Data is being collected from users in a batched / sequential manner. $\pi_\text{base}$ is updated to $\pi_\text{new}$ every time a batch of demonstrations is collected from a demonstrator.
>
> > if multiple operators are providing demonstrations in novel regions of the state space, aren’t they free to provide demonstrations that are incompatible with each other?
>
> Demonstrators giving contradictory demonstrations for novel states will never be a problem as we update  $\pi_\text{base}$  to $\pi_\text{new}$ after each demonstrator sequentially, a state that is novel for one operator ceases to be novel for a subsequent operator after retraining.
>
> >  I was not able to find the details of the likelihood and novelty computations in the main text or supplemental material: I’m assuming novelty is computed as the policy output variance across the ensemble of neural networks, but what does “likelihood” mean? Why does it only take the action as input, when likelihood seems to be a state-dependent measure (a particular action can be more likely at a given state than another one)? Furthermore, how exactly are the thresholds λ and η calculated, and what are the exact values they take in the experiments?
>
> We apologize that the detailed definition of novelty and likelihood was missing. We have edited the paper to provide a formal definition of these terms. Specifically, we use the thresholds detailed in the table below to compute the compatibility score $\mathcal{M}$ for the new demonstrations $D_\text{new}$. The likelihood of the action is measured using a negative mean squared error between the actions predicted by $\pi_\text{base}(s_{\text{new}})$ and the provided actions $a_{\text{new}}$. The action likelihood is a function of the current state and the provided action (as pointed out by R3). The novelty of a state is measured by the standard deviation in the predicted actions from the ensemble policy. We updated the paper with these details. To select these thresholds, we assume access to a compatible and an incompatible trajectory in addition to the base demonstrations. We regress these thresholds based on a 2D compatibility map of likelihood vs novelty (see Figure 2). We update our paper to make our assumptions as well as the definitions clear.

---

> > ### Author Response · Authors · 2022-08-25
> > **Last two days of the discussion period**
> >
> > With just a couple days left in the discussion period, we wanted to reach out to see if there are any final questions or clarifications we can provide!
> >
> > The key criticisms that Reviewer tzdZ pointed out were the lack of details in our method and some missing baselines in our active elicitation procedure; we believe that our added details about the experiments and baselines with filtering based on demonstration compatibility address this. We believe the remaining inline responses, extra experiments outlined in the general response, and revised draft (uploaded) also work to address any remaining concerns.
> >
> > We're excited to see how our reviewers view our paper in light of these changes. Again, if there are any remaining questions before the discussion period ends, please let us know. Thanks so much!

---

> > > ### Comment · Reviewer_tzdZ · 2022-08-25
> > > **One more clarification**
> > >
> > > Actually, one remaining question I have is why there are no Trajectory Length results for the food plating task in the Supplementary Table 4.

---

> > > > ### Author Response · Authors · 2022-08-26
> > > > **Updated Metrics for food plating**
> > > >
> > > > We have updated the pdf with updated metrics for trajectory collection to make Supplementary Table 4 more complete.
> > > > Please let us know if there are any remaining concerns/clarifications!
> > > > We appreciate your efforts in helping us improve the paper.

---

> ### Author Response · Authors · 2022-08-21
> **Response to Reviewer tzdZ (2/2)**
>
>
> > Line 212: an incompatible state-action pair is said to have M=0. Isn’t M a continuous measure that’s unlikely to be exactly 0?
>
> We thank the reviewer for noticing this typing error in defining M! We have changed the definition of M to the following to make the computation of compatibility clearer.
> \begin{equation*}
>     \mathcal{M} =
>     \begin{cases}
>         1 - \text{min}\left(\frac{ (\pi_{\text{base}}(s_{\text{new}}) - a_{\text{new}})^2}{\lambda}, 1\right) & \text{if novelty$(s_{\text{new}}) < \eta $}
>     \end{cases}
> \end{equation*}
> So, $ \mathcal{M}$ is 0 when $(\pi_{\text{base}}(s_{\text{new}}) - a_{\text{new}})^2 \geq \lambda$.
> > Line 216: The feedback phase uses state-similarity to retrieve relevant snippets of demonstrations. How is this done in image space? L2 pixel distance?
>
> For the retrieval of corrective demos, we look at the similarity of demos in the state space. This is done by measuring the L2 distance of the ResNet embeddings. Note that this isn't a perfect measure and that lots of other work tries to solve this problem; we choose ResNet features to be expedient. We add details to the supplementary.
>
> Please also see the general response for a list of revisions and how we address the concerns raised by the reviewers. Let us know if you have any other questions or concerns about the work that we could address.

---

> > ### Comment · Reviewer_tzdZ · 2022-08-25
> > **Response to Authors' Response**
> >
> > Thank you to the authors for their efforts in addressing my concerns. The revised manuscript is stronger as a result and I plan to improve my assessment to Weak Accept.

---

### Official Review · Reviewer_Esih · 2022-07-19

**Originality:** Good
**Technical Quality:** Fair
**Clarity Of Presentation:** Good
**Impact:** 3

**Recommendation:**

Weak Accept: I recommend accepting the paper, but will not argue for my recommendation if the majority of other reviewers have a different opinion.

**Summary:**

The paper proposes an interesting paradigm in which, to improve the consistency among the multiple demonstrations of a task, a measure of compatibility is used to guide the demonstrator. This guidance results in a consistent set of demonstrations, which in turn results in increased success rates. As a result, by eliminating the heterogeneity in the demonstration set, the learning is more efficient since multi-modal models are not required, and at the same time, tasks become more prone to succeed. The method was demonstrated in simulation on a simple task of inserting a nut onto a peg, and in a real robot on a task of placing food on a plate.

**Issues:**

One of the main issues with this paper is that it is easy to counterargue two of the main claims on which the contribution of the paper hinges on. First, that by suppressing heterogeneity the method could be potentially moving itself towards collecting a poor, local minima prone dataset of demonstrations. Many works try to do the opposite, which is to learn the richness of human variability via multi-modal models. Second, if the system is already so informed to the point it is capable of measuring the demonstration adequacy and providing feedback, then what is the actual reason to collect more demonstrations in the first place?

To improve upon the first issue, the paper could leverage insights from the literature on, for example, virtual fixtures and virtual guides [r1, r2]. In virtual guides, experiments motivate well the fact that although users may provide multi-modal demonstrations, it is clear that only a single mode is allowed, for example, in surgical tasks.
The tasks used in the current paper could be reformulated such that it becomes more evident the need for uni-modality. For example, a more convincing story could be written by changing the sentence from "This multimodality is inconsequential to human users, with task variations manifesting as subconscious choices; for example, reaching down, then across to grasp an object, versus reaching across, then down." to something along the lines "Although non-trained users may find natural to demonstrate a task in a variety of ways, there are certain tasks for which a single pattern is more preferable. Such patterns can be captured initially by expert demonstrations which are then used to provide the metric M which will guide subsequent users on how to demonstrate."

To improve upon the second, it would be useful to show why the data used to compute the likelihood in M is not enough to solve the task. For example, would it be possible to show that the current demonstrations to place the egg on the plate are too uncertain and thus more demonstrations are required? I think a look into active learning on robot demonstrations [e.g. r3] can provide some interesting insights to motivate the need for demonstrations.

Minor comment
======================
Under excessive correction, will users become less prone to demonstrate? Wouldn't it be better to let them demonstrate freely, and then use M to give single final feedback on the demonstration over the entire episode, rather than provide high-frequency feedback at each time step?

[r1] https://www.telerobotics.utah.edu/uploads/Main/Abbott_RoboticsResearch07.pdf
[r2] https://hal.archives-ouvertes.fr/hal-02158827/file/main_final.pdf
[r3] https://proceedings.mlr.press/v78/maeda17a.html

**Quality Of The Limitations Section:**

Limitations are addressed clearly

**Reviewer Expertise:**

4: The reviewer is confident but not absolutely certain that the evaluation is correct

**Robotics Focus:**

Sufficient demonstration on hardware

**Strengths And Weaknesses:**

The paper is clearly written and easy to follow. It has a straightforward structure. There are certainly merits of the idea of decreasing heterogeneity of the task and the paper motivates it quite well. Regarding the weaknesses, please, refer to the section on issues.

**Summary Of Recommendation:**

The experiments are too simple and do not seem to justify the complexity of the method. As it is, the narrative of the paper clashes with a body of literature on multi-modal learning but does not convincingly justify the paper's choice of approaching demonstration via loss of heterogeneity. I think there are many ways to improve the paper (see issues section) but this would require significant changes which are more appropriate for a re-submission.

---

> ### Author Response · Authors · 2022-08-21
> **Response to Reviewer Esih**
>
> We thank Reviewer `Esih` for their detailed read of our paper; we address their concerns around suppressing the multi-modality/heterogeneity of demonstrations, as well as the possibility of our active feedback suppressing user demonstration diversity.
>
> > suppressing heterogeneity the method could be potentially moving itself towards collecting a poor, local minima prone dataset of demonstrations.
>
> This is a great point; we argue that our work adds some benefit over the status quo in collecting demonstrations by allowing users to make *informed choices as to when to collect more demonstration data.*
>
> We strongly agree that users can have different preferences (multi-modality) in how the robot should perform certain tasks. However, for many “simple tasks/behaviors” (e.g., pick and place, transferring an egg from a pan to a plate in non-safety constrained settings), users may not care too much about how the robot does the task – just that the robot is able to succeed given minimal data.
>
> The underlying problem our work seeks to address is that without added information, users can collect “incompatible” demonstrations accidentally (i.e., demonstrations covering a novel mode) which leads to pathological failures when retraining the policy.
>
> Our contribution is just that our feedback mechanism *let’s the user know that they’re operating in a new mode!* If they would rather focus on getting the robot to successfully perform the task quickly as is often the case for most behaviors (e.g., the user is agnostic to how this happens), then our approach shows sizable gains by informing users about their demonstration compatibility with active feedback. Otherwise, users can make an informed choice about whether they want to collect new demonstrations reflecting their preferences – a win-win relative to the status quo!
>
> > Under excessive correction, will users become less prone to demonstrate? Wouldn't it be better to let them demonstrate freely, and then use M to give single final feedback on the demonstration over the entire episode
>
> This is an insightful point, and one that merited taking this week to get some new results!
>
> We ran new experiments to test a more realistic setting where a user might intervene with feedback at any point in execution rather than in the batched setting of our current experiments, **we apply our approach and run a user study using human-gated interactive corrections under the HG-DAgger framework.**
>
> Results: We don’t observe a reduction in user willingness to provide corrections, and actually show that with a slight interface design tweak to compensate for the real-time feedback, we don’t overload users either! **Our approach is able to scale to more interactive, real-time settings** – we believe this makes our paper significantly stronger, and only came to fruition as a result of the reviewer's insights!
>
>
>
> > Second, if the system is already so informed to the point it is capable of measuring the demonstration adequacy and providing feedback, then what is the actual reason to collect more demonstrations in the first place?
>
> We would like to clarify that we are considering a setting where the trained base policy is not good at performing the task reliably and fails to perform well in novel states (not covered in the base demos).[Something about how this is realistic and shows up pretty much in any IL scenario in today’s IL performance on various tasks] This is also reflected in the lower success rates for the base policy across the tasks that we study. We contend that measuring compatibility is much easier than generating the actions.
>
> Please also see the general response for a list of revisions and how we address the concerns raised by the reviewers. Let us know if you have any other questions or concerns about the work that we could address.

---

> > ### Author Response · Authors · 2022-08-25
> > **Last few days of the discussion period**
> >
> > With few days left in the discussion period, we wanted to reach out to see if there are any final questions or clarifications we can provide!
> >
> > A few of the key criticisms that Reviewer Esih was concerned about were the lack of discussion of unimodality vs multimodality in demonstrations, and why using the base policy for feedback made sense; we believe that our added discussions on the relation of multimodality to active elicitation and the capabilities of the base policy address these concerns. The reviewer was also concerned about  the high frequency of feedback to the users, we added a user study with HG DAgger and active elicitation to address this. We believe the remaining inline responses, extra experiments outlined in the general response, and revised draft (uploaded) also work to address any remaining concerns.
> >
> > We're excited to see how our reviewers view our paper in light of these changes. Again, if there are any remaining questions before the discussion period ends, please let us know. Thanks so much!

---

### Official Review · Reviewer_hqZQ · 2022-07-31

**Originality:** Very Good
**Technical Quality:** Very Good
**Clarity Of Presentation:** Excellent
**Impact:** 3

**Recommendation:**

Weak Accept: I recommend accepting the paper, but will not argue for my recommendation if the majority of other reviewers have a different opinion.

**Summary:**

This paper proposes an interactive method to inglobate new demonstrations into an existing dataset and policy in such a way that it is not detrimental to the existing policy. The authors propose to do so by measuring how similar and compatible the new demonstration is with the existing policy, and discarding it in case the algorithm finds it too dissimilar. They also extend this technique to provide online feedback to a human operator on how to better record the demonstration.
The method is based on the assumption that, in a state already visited, the new demonstration should be close to the existing policy. In a novel state, however, new data is always good. The algorithm therefore computes at each step a novelty and consistency measure based on the existing policy and dataset.

**Issues:**

No particular issue. I would suggest for future extensions to also test it on more challenging, longer horizon tasks.

**Quality Of The Limitations Section:**

Limitations are addressed clearly

**Reviewer Expertise:**

3: The reviewer is fairly confident that the evaluation is correct

**Robotics Focus:**

Sufficient demonstration on hardware

**Strengths And Weaknesses:**

I believe the problem this paper tackles is important, and methods to have interactive learning between robots and human may be beneficial. Although in the future we may want to learn complex policies from heterogeneous data, at the moment robot learning can't scale datawise as other fields, and therefore these techniques are still fundamental.
Strengths:
- Interesting problem, tackled with a simple but well designed approach.
- The experiments section is well written, and indeed shows the effect of an incompatible additional dataset, showing that in this scenarios more data is not always good. Real robot experiments also further prove the effectiveness of the algorithm.
- The interactive algotithm they propose is interesting and potentially helpful especially at larger scale.
Weaknesses:
- Tasks are still relatively simple and do not involve the use of longer horizon skills.

**Summary Of Recommendation:**

I suggest to accept the paper. The method presented is intuitive but effective and can be beneficial also in larger scale settings. While the tasks do not show particularly complex skills, I believe they still prove the improvements that the algorithm brings, also in the real world.

---

> ### Author Response · Authors · 2022-08-21
> **Response to Reviewer hqZQ**
>
> We appreciate Reviewer `hqZQ` for their thorough read and positive review of our work; we address their concerns around the complexity of tasks here.
>
> > Tasks are still relatively simple and do not involve the use of longer horizon skills.
>
> We argue that our tasks – especially the `Hammer` tool-use task, and the `Peg-Insertion` tasks – are long-horizon. We would like to argue that our sim tasks, in particular, hammer placement (introduced in [Bottom-Up Skill Discovery from Unsegmented Demonstrations for Long-Horizon Robot Manipulation](https://arxiv.org/abs/2109.13841)) is a complex task requiring multiple skills combined over a long horizon: navigating to different objects, opening a drawer, picking up the hammer, placing the hammer, and closing the drawer. Succeeding at this task without demonstrations and human feedback would require significant reward engineering.
>
> Moreover, the tasks presented in our work validate the key idea of our approach; adding even more complex tasks would have limited contributions towards proving the validity of our method. The key idea is that our compatibility measure and active elicitation procedure are meant to be *general* components that *can be used to constrain the space of possible policies, which can then be used to guide learning for arbitrary tasks.*
>
> Please also see the general response above for a list of revisions and how we address the concerns raised by the reviewers. Let us know if you have any other questions or concerns about the work that we could address.

---

> > ### Author Response · Authors · 2022-08-25
> > **Last few days of the discussion period**
> >
> > With just few days left in the discussion period, we wanted to reach out to see if there are any final questions or clarifications we can provide!
> >
> > One of the key criticisms that Reviewer hqZQ pointed out was the lack of complex long horizon tasks in our evaluations; we believe that our argument of why Nut Assembly and Hammer Placement are representative complex long horizon tasks addresses this. We believe the remaining inline responses, extra experiments outlined in the general response, and revised draft (uploaded) also work to address any remaining concerns.
> >
> > We're excited to see how our reviewers view our paper in light of these changes. Again, if there are any remaining questions before the discussion period ends, please let us know. Thanks so much!

---

### Author Response · Authors · 2022-08-21
**General response and revised files**

**Comment:**

It’s wonderful to see the positive response around our work – from Reviewer `hqZQ`’s note that this is an “[i]interesting problem, tackled with a simple but well-designed approach,” to Reviewer `tzdZ`’s opinion that our “high-level approach is compelling and novel.” We’re grateful that all our reviewers and meta-reviewer found our paper “very well-written, clear, and easy to follow.”

We are also appreciative of their constructive feedback, questions, and concerns; to fully address everything, we have **added new experimental results, updated the main text and supplemental with additional discussion, and added clarity/additional arguments to answer specific concerns**. Specifically, we have added:

1. **New Experiments**
- [Re: “Lack of ablations around the active elicitation procedure”; `tzdZ`, Meta-Reviewer]:
Added new experiments (**Section 5.1 in uploaded revision**) that filter demonstrations collected under the “naive” data collection with our defined compatibility measure (as suggested by Meta-Reviewer). We show that this “passive” filtering improves over the baseline condition (no-feedback), while active feedback performs better by a large margin.
- [Re: “High frequency feedback discouraging corrections from users (our approach is excessive and limits the diversity/quality of demonstrations)”; `Esih`]:
We ran a new experiment (**Appendix F**) in which users are able to intervene within a demonstration via Hg-DAgger ([“HG-DAgger: Interactive Imitation Learning with Human Experts”](https://arxiv.org/abs/1810.02890)) with feedback isolated to certain sub-sections. We also add new evaluation metrics specifically around active elicitation (**Appendix G**).
The results of this more granular active feedback/response protocol show that we’re not discouraging users, **demonstrating that our approach can scale to more interactive, “real-world” collection scenarios**(a la work like [SayCan](https://arxiv.org/abs/2204.01691)).

- [Re: “Extra ablations, generality of approach to different horizon tasks, complex scenarios”; `hqZq`, `tzdZ`, Meta-Reviewer]:
We add a BC-RNN baseline (**Appendix C**) to test more complex policy classes showing that active elicitation remains performant even as task success grows, and new evaluation metrics with a thorough discussion (**Section 5.1**). In the individual response to Reviewer `hqZQ` we discuss our `long-horizon` tasks and how this relates to our approach.


2. **Higher-Level Concerns & Questions**
- [Re: “users naturally provide heterogeneous/multi-modal demonstrations, while our approach tries to elicit demonstrations in a “single” (possibly sub-optimal) mode”; `Esih`]:
We provide thorough arguments in the individual response to Reviewer `Esih` as well as an updated discussion in (**Section 6**).  Crisply, we agree that we do not want to suppress multi-modal demonstrations; however, for simple functional tasks (such as pick and place, opening a bottle, opening/closing drawers or doors) where users don’t have strong preferences just want the robot to succeed, our approach is able to accelerate policy learning.

&emsp;&emsp;In cases where users **do** have strong preferences, we are instead able to allow users to make an informed choice about whether they want to spend the time collecting more demonstrations to reflect their preferences. Compared to the status quo where users just don’t know if their demonstrations are incompatible in the first place, our approach provides them with more feedback to make an informed choice when it comes to demonstration collection.


3. **Clarifications**: We add additional details for the compatibility metric and corresponding likelihood computation (**Section 4.1**), details around the sequential updates of the base policy given incoming user demonstrations (**Section 5**), details on how we compute the threshold for compatibility (**Appendix B**), and qualitative details (**Appendix D, E**) to enrich our evaluation.

We hope that our revisions and more detailed individual responses below cover our reviewers’ outstanding concerns, and hope that our reviewers might consider raising their scores. If there are any remaining concerns, we look forward to addressing them over the rest of the rebuttal period! Thank you all!



**Zip File:**

/attachment/320637d1b853b0082a23487281532af1d6307a68.zip

---

### Meta-Review · Area_Chair_rsiH · 2022-08-14

**Recommendation:** Accept (Poster)
**Confidence:** 4

**Metareview:**

### Strengths
- The paper tackles a very interesting problem with a sound method.
- Well written experimental section
- Real robot experiments and user study

### Weaknesses
- Showing that the method also works on more complex tasks would have been nice
- Enforcing consistency (rather than allowing multi-modality) needs to be motivated better
- Why being able to measure demonstration adequacy isn't sufficient by itself needs a better motivation, and should ideally also be shown experimentally
- Some important details need to be added

### Summary
A promising paper that needs some additional explanations and (ideally) experiments.

### After rebuttal and discussion
All reviewers replied and/or participated in the reviewer&AC discussion. The replies managed to clear up quite a few doubts. While not groundbreaking - in terms of novelty of the way of querying for demonstrations and experimental results - this is a solid, well-written paper tackling a relevant problem

**Best Paper Nomination:**

No